

Technical note: Comparison between two generalized Nash models
with a non-zero initial condition
Baowei Yan[*], Aoyu Yuan, Zhengkun Li, Chen Cao
*School of Hydropower and Information Engineering, Huazhong University of Science and*
*Technology, Wuhan, China*
**Abstract** Initial condition can impact the forecast precision especially in a real-time
forecasting stage. The discrete linear cascade model (DLCM) and the generalized Nash
model (GNM), expressed in different ways, are both the generalization of the Nash
cascade model considering the initial condition. This paper investigates the relationship
and difference between DLCM and GNM both mathematically and experimentally.
Mathematically, the main difference lies in the way to estimate the initial storage state.
In the case of n=1, it was shown theoretically that the difference between the two
models is whether the current outflow is estimated (DLCM) or observed (GNM). The
GNM is the unique solution of the Nash cascade model with a non-zero initial condition,
while the DLCM is an approximate solution and it can be transformed to the GNM
when the initial storage state is calculated by the approach suggested in the GNM. At
last, a test example obtained by the solution of the Saint-Venant equations is used to
illustrate this difference. The results show that the GNM provides a unique solution
while the DLCM has multiple solutions depending on the estimate accuracy of the
current state.

---

[*] Corresponding author. Tel.: +86 27 87543992; fax: +86 27 87543992
E-mail address: bwyan@hust.edu.cn (B. Yan)



**Keywords** generalized Nash model; discrete linear cascade model; initial storage state;
unique solution
1. **Introduction**

24        In hydrology, the concept of linear reservoir cascade suggested by Nash (1957) is

widely used in connection with the mathematical modeling of surface runoff. Several
Nash cascade based models have been developed to model the rainfall runoff process
and river flow routing (Yan et al., 2015). In the original linear reservoir cascade model,
the initial storage of each reservoir is assumed to be zero, or equivalently the reservoirs
are empty. The initial state is usually thought to be insignificant in the forecasting as its
effect will vanish after a sufficiently long simulation time. But for some short time
prediction situations, just like the identification of the impulse-response function and
the real-time forecasting, the initial state will produce relatively great impact. Szollosi-
Nagy (1982) formulated a state-space description of the Nash cascade model i.e. the
discrete linear cascade model (DLCM) in a matrix form whereby the initial state was
included that can be thought of a generalization of the Nash cascade model. The
determination of the initial state of the DLCM was then proposed by Szollosi-Nagy
(1987) via observability analysis. The DLCM was discretized originally in a pulse-data
system framework which seems more suitable for the irregularly changing precipitation
but not necessarily for the gradually changing streamflow. Under a linear change
assumption of the input, the DLCM was extended by Szilagyi (2003) to a sample-data
system framework. Since then, Szilagyi and his team have made great effort to develop
this model (Szilagyi, 2006; Szilagyi and Laurinyecz, 2014). With so many advantages



that have been summarized by Szilagyi (2006), the DLCM has been in operational use
for over 30 years in Hungary. However, it has not yet been applied more broadly except
in Hungary, or furthermore by Szilagyi and his team. One possible reason may be due
to the complicated mathematical expression and calculation. The development of a
simpler expression of the DLCM is necessary to make it more popular and applicable
in practice.
Recently, Yan et al. (2015) published a paper "The generalized Nash model for river
flow routing" in Journal of Hydrology. In that paper, the Laplace transform and the
principle of mathematical induction were used to solve the $n$th order nonhomogeneous
linear ordinary differential equation (NLODE) of the Nash cascade model with a non-
zero initial condition. The generalized Nash model (GNM), i.e. the analytical solution,
with a simpler expression of the Nash cascade model was obtained. What's more, the
GNM has been physically interpreted, which makes it to be a conceptual model and not
only a mathematical formulation. The DLCM was also obtained from the Nash cascade
model with the same initial condition. But whether the expressions or the simulation
results of these two models are differently exhibited. There may be some confusions to
the model users. It is necessary to distinguish these two models for the users. Hence,
this paper try to investigate the relationship and difference between DLCM and GNM
both mathematically and experimentally.
2. **Relationship between the DLCM and the GNM**
In the derivation of the DLCM, a state-space matrix approach was used. The state
and output equations of the Nash cascade model are formulated as follows (Szollosi-



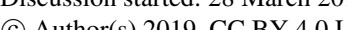



Nagy, 1982; Szilagyi, 2003)

$$\mathbf{S}(t) = \mathbf{\Phi}(t, t_0)\mathbf{S}(t_0) + \int_{t_0}^{t} \mathbf{\Phi}(t, \tau)\mathbf{G}I(\tau)d\tau \qquad (1)$$

$$O(t) = \mathbf{H}\mathbf{S}(t) \qquad (2)$$

where $\mathbf{S}(t)$ is the storage state vector and denotes the stored water volumes of the $n$
linear cascade reservoirs,

$$\mathbf{\Phi}(t, t_0) = \begin{bmatrix} e^{-\frac{t-t_0}{K}} & 0 & \cdots & 0 \\ \frac{t-t_0}{K}e^{-\frac{t-t_0}{K}} & e^{-\frac{t-t_0}{K}} & \cdots & \vdots \\ \vdots & \ddots & \ddots & 0 \\ \frac{(t-t_0)^{n-1}}{K^{n-1}(n-1)!}e^{-\frac{t-t_0}{K}} & \cdots & \frac{t-t_0}{K}e^{-\frac{t-t_0}{K}} & e^{-\frac{t-t_0}{K}} \end{bmatrix}. \qquad (3)$$

is the state transition matrix, $t_0$ is the initial time, $K$ is the storage coefficient,
$\mathbf{G}=[1,0,\ldots,0]^{\mathrm{T}}$, $I(.)$ is the instantaneous inflow of the first reservoir, $O(t)$ is the
outflow, and $\mathbf{H}=[0,0,\ldots,1/K]$.

Combining Eqs. (1) and (2) and assuming $t_0=0$, one obtains (Szilagyi, 2006)

$$O(t) = \mathbf{H}\mathbf{\Phi}(t,0)\mathbf{S}(0) + \int_{0}^{t} u(t-\tau)I(\tau)d\tau \qquad (4)$$

where $u(.)$ is the instantaneous unit hydrograph.

Eq. (4) is the basic formula for DLCM. For the discrete streamflow data system,

assuming that both input and output are sampled at equidistant sampling intervals $\Delta t$,
the recursive form of the DLCM can be written as follows (Szilagyi, 2006)

$$O(t+\Delta t) = \mathbf{H}\mathbf{\Phi}(\Delta t,0)\mathbf{S}(t) + \int_{t}^{t+\Delta t} u(t+\Delta t-\tau)I(\tau)d\tau \qquad (5)$$

Once the initial storage state vector $\mathbf{S}(0)$ is obtained, the outflow can be estimated

recursively by using Eqs. (1) and (5). The identification of $\mathbf{S}(0)$ is an inverse problem,
which can be computed by inverting Eq. (4) and from the first n input-output pairs, as




originally proposed by Szollosi-Nagy (1987) and Szilagyi (2003), i.e.

$$\mathbf{S}(0) = \mathbf{\Omega}_n^{-1} \left( \varepsilon_1, \varepsilon_2, \cdots, \varepsilon_n \right)^T \tag{6}$$

where

$$\mathbf{\Omega}_n = \left[ \mathbf{H\Phi}(\Delta t, 0), \mathbf{H\Phi}^2(\Delta t, 0), \cdots, \mathbf{H\Phi}^n(\Delta t, 0) \right]^T \tag{7}$$

and

$$\varepsilon_i = O(i\Delta t) - \int_0^{i\Delta t} u(t-\tau)I(\tau)d\tau, \ i=1,\cdots,n. \tag{8}$$

That's the complete procedure of the DLCM. It is a discrete solution of the Nash
cascade model. Actually, the initial storage state vector $\mathbf{S}(0)$ can be calculated by
another simpler approach that has been proposed by Yan et al. (2015). From the linear
storage-outflow relationship suggested in the Nash cascade model, we have

$$\mathbf{S}(0) = \left[ KO_1(0), KO_2(0), \cdots, KO_n(0) \right]^T \tag{9}$$

where   $O_j(0)$ ( $j=1,\cdots,n$ )  is the initial outflow of the $j$th reservoir and can be
computed by (Yan et al., 2015)

$$O_j(0) = \sum_{i=0}^{n-j} C_{n-j}^i K^i O^{(i)}(0) \tag{10}$$

where   $O^{(i)}(0)$  is the $i$th derivative of  $O(0)$ , and

$$C_n^r = \frac{n!}{r!(n-r)!} \tag{11}$$

is the combination formula. Note that   $O_n(0)$   is equal to the initial downstream outflow
$O(0)$ . Then

$$\mathbf{H\Phi}(t,0)\mathbf{S}(0) = \sum_{j=1}^{n} \frac{e^{-\frac{t}{K}}}{(n-j)!} (\frac{t}{K})^{n-j} \sum_{i=0}^{n-j} C_{n-j}^i K^i O^{(i)}(0)$$

$$= \sum_{j=0}^{n-1} \frac{e^{-\frac{t}{K}}}{j!} (\frac{t}{K})^j \sum_{i=0}^{j} C_j^i K^i O^{(i)}(0) \tag{12}$$





Note that  $\mathbf{H\Phi}(t,0)\mathbf{S}(0)$  is just equal to $R_0(t)$ that has been defined in the GNM (Yan
et al., 2015). Substituting Eqs. (3) and (9) into Eq. (4) gives

$$O(t) = e^{-\frac{t}{K}} \sum_{j=0}^{n-1} \sum_{i=0}^{j} O^{(i)}(0) \frac{t^j}{i!(j-i)!K^{j-i}} + \int_0^t u(t-\tau)I(\tau)d\tau \qquad (13)$$

That's just the GNM that has been proposed by Yan et al. (2015). Hence, the DLCM
can be transformed to the GNM when the initial storage state is calculated by the linear
storage-outflow relationship.
3. **Difference between the DLCM and the GNM**

The main difference between the two models lies in the estimation of the initial

storage state. In the DLCM, the initial storage state $\mathbf{S}(0)$ is expressed as a function of
the first n input-output pairs, while in the GNM, it is expressed as a function of the $i$th
derivative of the initial outflow.

To further illustrate this difference, take the special case of $n=1$ as an example. In

the DLCM, the initial storage $S(0)$ can be estimated by (Szollosi-Nagy, 1987)

$$S(0) = Ke^{\frac{\Delta t}{K}} \left[ O(\Delta t) - (1 - e^{-\frac{\Delta t}{K}})I(0) \right] \qquad (14)$$

It is suggested that the initial storage is estimated by the input/output pair [$I(0)$,

$O(\Delta t)$]. In fact, when $n=1$, the river flow routing system can be described by the
following NLODE (Szollosi-Nagy, 1982; Yan et al., 2015)

$$KO^{'}(t) = I(t) - O(t) \qquad (15)$$

It is easy to get the solution of this NLODE with a result of

$$O(t) = O(0)e^{-\frac{t}{K}} + \int_0^t u(t-\tau)I(\tau)d\tau. \qquad (16)$$

Provided that $I(t)$ is taken to be constant at the value it obtains at time $t$, in the [$t$, $t+$





$\Delta t$ ] interval (Szilagyi, 2003), for one step ahead, we obtain

$$O(\Delta t) = O(0)e^{-\frac{\Delta t}{K}} + (1 - e^{-\frac{\Delta t}{K}})I(0) \qquad (17)$$

Then the initial outflow can be estimated by

$$O_{est}(0) = e^{\frac{\Delta t}{K}} \left[ O(\Delta t) - (1 - e^{-\frac{\Delta t}{K}})I(0) \right] \qquad (18)$$

where $O_{est}(0)$ is the estimated initial outflow.
Combing equations (14) and (18), yields

$$S(0) = KO_{est}(0) \qquad (19)$$

On contrary, in the GNM, the initial state is directly obtained by equation (9) based
on the concept of linear reservoir with a result of

$$S(0) = KO_{obs}(0) \qquad (20)$$

where $O_{obs}(0)$ is the observed initial outflow.
Comparison of Eqs. (19) and (20) shows that the difference between the two models
in the case of $n=1$ lies in the fact that whether the initial outflow is estimated or observed.
In the DLCM, the initial outflow $O(0)$ is estimated by the observed input/output pair
$[I(0),\ O(\Delta t)]$, as shown in Eq. (18). In fact, at initial time, the outflow for the next time
step $O(\Delta t)$ is still unknown, while the observed initial outflow $O(0)$ is available at
that time and doesn't need to be estimated. Instead, this observed value is directly used
in the GNM. Theoretically, they are equivalent with the same numerical values if no
predict error exists. However, the predict error is virtually inevitable. Though this error
may be ignored in some cases, it's at least a truth for a real-time forecasting that the
estimation of the current state $\mathbf{S}(t)$ depends on the current inflow $I(t)$ and the outflow
for the next time step $O(t + \Delta t)$ when the recursive DLCM is employed according to





Eq. (5). It seems paradoxical because the outflow for the next time step $O(t+\Delta t)$ is
still unknown at the current time and is to be predicted by using the current state $\mathbf{S}(t)$.
The approach used in the DLCM to deal with this paradox is to estimate the current
state $\mathbf{S}(t)$ by applying the transition matrix to the initial state from Eq. (1). In the case
of n=1, $\mathbf{S}(t)$ calculated from Eq. (1) can be simplified as follows
$$S(t) = S(0)e^{-\frac{t}{K}} + \int_0^t e^{-\frac{t-\tau}{K}} I(\tau)d\tau \tag{21}$$

where $S(0)$ is estimated by Eq. (14). Then the recursive DLCM can be written as
$$O(t+\Delta t) = \frac{1}{K}S(t+\Delta t)$$

$$= \frac{1}{K}\left[ S(t)e^{-\frac{\Delta t}{K}} + \int_t^{t+\Delta t} e^{-\frac{t+\Delta t-\tau}{K}} I(\tau)d\tau \right]$$

$$= \frac{1}{K}S(t)e^{-\frac{\Delta t}{K}} + (1-e^{-\frac{\Delta t}{K}})I(t). \tag{22}$$

It is suggested from Eq. (21) that the current state $S(t)$ depends to some extent on
the initial state $S(0)$, or equivalently, the current state is not unique since any time before
current time can be taken as the initial time. As a result, the outflow for the next time
step $O(t+\Delta t)$ determined by $S(t)$ and $I(t)$ from Eq. (22) will have multiple solutions.
While in the recursive GNM, the current state is uniquely determined by the current
outflow $O(t)$ according to Eq. (9) in which the initial time is set to the current time. In
this case, the recursive GNM has the following unique expression
$$O(t+\Delta t) = O(t)e^{-\frac{\Delta t}{K}} + (1-e^{-\frac{\Delta t}{K}})I(t) \tag{23}$$

Comparison of Eqs. (22) and (23) suggests that the only difference between the
recursive form of the two models in the case of $n=1$ lies in whether the current outflow
is estimated or observed. Similarly, for $n>1$, the current outflow of the last reservoir



(i.e. $n$th reservoir) in the Nash model is also estimated rather than observed in the
DLCM. Hence, the DLCM is an approximate solution but not the exact solution of the
Nash cascade model. As an analytical solution, the GNM is applicable to the natural
continuous streamflow system. However, in practice, the streamflow data are usually
discretely measured. The derivative term in the GNM doesn't exist in the discrete
streamflow data system. To make the GNM practically applicable, the numerical
calculation approach such as the finite difference method is often used. While the
DLCM, as a discrete solution, can be directly applied to the discrete streamflow data
system.
4. **An illustrative example**
A test example was used to further illustrate this difference. This example was
obtained by numerically integrating the Saint-Venant equations of open channel flow
over a rectangular channel of $L$=120 km in length, $B = 20$ m in width and a constant
channel slope $S_0 = 0.0002$. The Manning's roughness parameter $n_0$ was set to 0.004 for
the entire length of the channel. The upstream boundary condition was defined by the
following inflow hydrograph (Camacho and Lees, 1999)
$$I(t) = I_b + (I_p - I_b)\left(\frac{t}{t_p}\right)^{\frac{1}{\gamma-1}} \exp\left(\frac{1-t/t_p}{\gamma-1}\right) \tag{23}$$

where $I_b$ is the initial steady flow (100 m³/ s) in the reach; $I_p$ is the peak flow (300 m³/
s); $t_p$ is the time to peak (20 h) and $\gamma$ is the skewness factor (1.2). The downstream
boundary condition, fixed at 120 km downstream, was defined by a looped-rating curve
based on the Manning equation for normal flow.



The hydrograph was routed to distances of 20, 40, 60, 80, 100 and 120 km from the
inflow section. To minimize somewhat artificial nature of the upper and lower boundary
conditions (Szilagyi, 2006), the middle reach between 40 km and 80 km was selected
for flow routing, i.e. the flowrate values given by the Saint-Venant equations at 40 km
and 80 km served as the "observed" upstream and downstream flow values, respectively.
The SCE-UA global optimization algorithm (Duan et al., 1994) was used to optimize
parameters in the two models by directly minimizing the root mean squared error, with
same optimized values of $n = 1$ and $K = 4.6$ h. In the real-time forecasting, for example,
take $t=16$ h as the current time, then any time before $t=16$ h can be taken as the initial
time $t_0$ in the DLCM. If $t_0 = 1$ h, the current state can be estimated by combing Eq. (14)
and (21), and further the current outflow, i.e. $O$ ($t=16$ h) can be calculated by linear
storage-outflow relationship, with a result of 126.13 $m^3/s$. If $t_0 = 15$ h, the current
outflow $O$ ($t=16$ h) was estimated by the same procedure with a result of 118.58 $m^3/s$.
Then this value was used to estimate the following outflow by using the Eq. (22). While
for the GNM, the "observed" value of $O$ ($t=16$ h) $=113.92$ $m^3/s$ was directly used to
estimate the outflow recursively by using the Eq. (23). The hydrographs obtained by
the DLCM with different initial time as well as the GNM were illustrated in Fig. 1.
With different current outflow, the DLCM correspondingly provided different
forecasted discharge values, especially the first few ones. The Nash–Sutcliffe efficiency
coefficient ($E_{NS}$) values for $t_0 = 1$ h and $t_0 = 15$ h were 0.9882 and 0.9919, respectively.
The GNM provided the unique and also the best forecasted results, with a result of $E_{NS}$
$= 0.9928$. It is shown from this example that the DLCM has multiple solutions and the

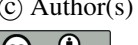


forecast precision depends upon the estimate accuracy of the current state.

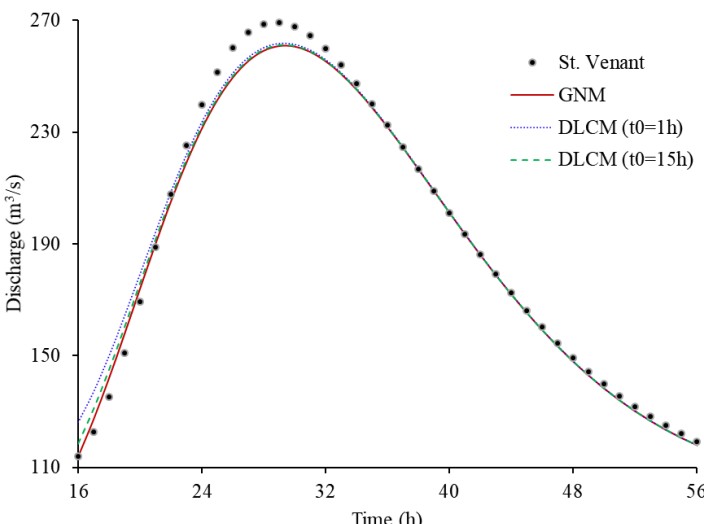


Figure 1. Routing results for the DLCM and the GNM

**5. Conclusion**

The DLCM formulated the continuous Nash cascade model in a matrix form based

on the principles of state space analysis. The identification of the initial state is required
when performing the DLCM. There is a paradox in the identification process that the
outflow at the next time step used to estimate the initial storage is still unknown at initial
time. To deal with this paradox in the real-time forecasting stage, the current state is
estimated by applying the transition matrix to the initial state. Due to the nonuniqueness
of the initial time, the DLCM will have multiple solutions. So the DLCM is an
approximate solution of the Nash cascade model but not the exact solution. The GNM
has been derived theoretically by solving the $n$th order NLODE of the Nash routing
theory. It's the unique analytical solution of the Nash cascade model. With an analytical
expression, the initial state is implicitly written in a form of derivative, and it does not



need to be estimated separately. Besides, the DLCM can be transformed to the GNM
when the initial storage state is calculated by the approach used in the GNM.
**Acknowledgments.** This study is financially supported by the National Key R&D
Program of China (2016YFC0402708), and the Fundamental Research Funds for the
Central Universities (HUST: 2017KFYXJJ195, 2016YXZD048).

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
