# Peer review of "Technical note: Comparison between two generalized Nash models"

_Hydrology and Earth System Sciences, 2019_

## Short Comment (SC1) · 2 Apr 2019

The authors compare apples with oranges. They derive a solution for a continuous-time system where the inflow rate is an analytical function. The Discrete Linear Cascade Model they compare their solution to, on the other hand, is for a system where the inflow rate is sampled at discrete time intervals (dt). This means that the inflow rate is available as instantaneous values only, separated by dt intervals. Between the instantaneous measurements, the inflow rate is either assumed (as no information is available) to keep the last measured value (i.e., pulsed data-system) or is considered as linearly changing between subsequent measurements. The solutions of the DLCM, discussed

in length in Szilagyi-Szollosi-Nagy (2010) therefore are the exact solutions under the practical conditions of discrete sampling of a continuous variable/signal (i.e., stream-flow rate). The authors' claim therefore that the DLCM solutions are only approximate is erroneous and misleading and shows a complete lack of understanding the difference between a continuous and a discretely sampled signal/system. While the DLCM solutions were worked out for the specific practical situation of discretely sampled flow rates, encountered at any hydrological forecasting service (such as the National Hydrological Forecasting Service of Hungary, where the model has been in operational use for more than 30 years), the analytical solution the authors derive is useless for such purposes, as the different-order time-derivatives required for their solution are simply non-existent for such discretely sampled signals, made up of piece-wise straight [joint or disconnected (the latter for pulsed data)] line-segments. I have explained this for the authors several times before as a reviewer of their manuscript that they had submitted to HESSD this time.

References: Szilagyi J., Szollosi-Nagy, A. (2010). Recursive streamflow forecasting: a state-space approach, CRC Press, Taylor & Francis, Boca Raton, FL, USA, pp. 195, ISBN 978-0-415-56901-9.

---

## Author Comment (AC1) · 8 Apr 2019

We would like to thank J. Szilagyi for his comments on our manuscript. The main ideas of Szilagyi are that the DLCM is the exact solution of the discrete system, and the GNM is the analytical solution of the continuous system which is useless in practical as the derivatives required are non-existent. So he think there is no need to compare these two models. Our responses are as follows. (1) Both the DLCM and GNM are derived from the Nash cascade model with a non-zero initial condition. Theoretically, such problem with the same initial condition should have a unique solution. But whether the expressions or the simulation results of these two models are differently exhibited.

This may confuse the model users. It is necessary to distinguish these two models for the users. The main purpose of this manuscript is to clarify the relationship and difference between these two models. (2) Szilagyi repeatedly argued the DLCM is a solution of the so called discrete system. In fact, the river flow system is originally a continuous natural system though the streamflow are rarely measured continuously in time in practice restricted to the sampling technique. The discrete streamflow data are just an approximation of the natural river system. The theories or models built based on the continuous natural system, e.g. the Saint Venant equations and the Nash cascade model, are certainly valid to the discrete data system. For the discrete streamflow data system, the derivative term in the GNM doesn't exist, but it can be numerically calculated by using the finite difference method which can be seen in the reference of Yan et al. (2015). Hence, the GNM derived from the continuous Nash cascade model is certainly applicable to the discrete data system. The simulated results of the illustrative example have also proved this. (3) One key issue Szilagyi skirted round is the identification of the initial state which is also the main difference between these two models. How to estimate the initial state determines the solution is exact or approximate. In the DLCM, the identification of initial state is treated as an inverse problem, which also means that the state at the end of time step is initially available. It seems paradoxical because the state at the end of time step is still unknown at the initial time. As a result, the initial state in the DLCM is an approximate value. Therefore, even for a discrete system, the DLCM is still an approximate solution of the Nash cascade model. But for the GNM, as interpreted in the manuscript, the initial state is strictly calculated by the linear storage-outflow relationship suggested in the Nash cascade model. It is only depending on the initial value of the outflow. The GNM is mathematically derived from the nth order nonhomogeneous linear ordinary differential equation of the Nash cascade model. It is the unique exact solution of the Nash cascade model.

---

## Referee Comment (RC1) · Anonymous Referee #1 · 1 Jul 2019

This technical notes compares two solutions to a Nash cascade model. The note is generally well-written but content is highly mathematical and probably only accessible to a small number of people. This is of course not necessarily a problem, but in this case I am struggling to understand what is the exact nature of the problem, and how the proposed work constitutes an improvement. It also appear that the model has previously been published in a recent J. Hydrology paper.

The improvements seems only to be only very minor as reported in Section 4, and the comparison of the hydrographs in Figure 1 are almost indistinguishable. Finally, the conclusion is a general summary, and not really related to the results presented in the

note - it could have been written before anything else in the note.

---

## Referee Comment (RC2) · Anonymous Referee #2 · 24 Sep 2019

The study is based on the comparison of two well-known models It seems that the objective is truly very narrow and does not contribute significantly to enhancing our understanding of hydrological system functioning. As such, I find it of very limited relevance to HESS. It might be more suitable to an application- or mathematically-oriented Journal, even as I am not sure that there are significant advancements in terms of mathematical developments.

A point which is not entirely clear to me is the basis for statements of the kind "The results show that the GNM provides a unique solution while the DLCM has multiple solutions depending on the estimate accuracy of the current state.". They then state that

observed values do not need to be estimated, thus implying that observed values are not associated with uncertainties. If a key difference is related to GNM being associated with observations (and not estimates), one could also claim that observations are always associated with measurement uncertainty/error. How do the Authors reconcile this aspect? There seem to be no mention of this aspect in the study.

In the example section, the Authors mention relying on an optimization approach to estimate model parameters. It seems to me that parameter estimation uncertainty is neither quantified nor considered and I am not sure why.

With reference to non-uniqueness of the solution, I am not sure why this is not compatible with typical uncertainty propagation analyses that are performed in environmental systems. Since there is uncertainty in some quantities, the latter should propagate to model outputs. Such an uncertainty can also be associated with initial conditions. The Authors should also comment on these aspects in future works.

In terms of comparisons, I am not sure about the point raised by the Authors. They claim that the results obtained by the DLCM approach are approximated (I guess when considering results obtained through the continuous counterpart of an otherwise discretely sampled signal). I am not sure about what elements we learn from this exercise with respect to other studies on signal analysis that are available in the literature.

Finally, it should be noted that the quality of the English is really substandard, thus posing difficulties to the reviewer. I am not providing specific examples simply because they are widespread throughout the text.

---

## Author Comment (AC2) · 21 Oct 2019

**Responses to Anonymous Referee #1**

This technical notes compares two solutions to a Nash cascade model. The note is generally well-written but content is highly mathematical and probably only accessible to a small number of people. This is of course not necessarily a problem, but in this case I am struggling to understand what is the exact nature of the problem, and how the proposed work constitutes an improvement. It also appear that the model has previously been published in a recent J. Hydrology paper.

Reply: We thank Referee #1 for his/her positive evaluations. In the paper published in J. Hydrology, the detailed theoretical derivation of the Generalized Nash Model (GNM) was made. In our recent research, we found that the discrete linear cascade model (DLCM) proposed by Szollosi-Nagy and then developed by Szilagyi is similar to GNM. These two models are both derived from a same problem, i.e., the Nash cascade model with a non-zero initial condition. Theoretically, these two models should have a same expression and a same result. However, whether the expressions or the simulation results of these two models are differently exhibited. This may confuse the model users. As the proposer of the GNM, we should have the responsibility to clarify these confusions. In this manuscript, the relationship between these two models was first investigated by the reconstruction of the DLCM. Then the difference was found in the process of the reconstruction. This comparison can help the model users make a clearer understanding of these two models. Furthermore, along with the reconstruction of the DLCM, the interpretation of the DLCM was also made, which makes it more conceptual in hydrology and not only a mathematical formulation any more.

The improvements seems only to be only very minor as reported in Section 4, and the comparison of the hydrographs in Figure 1 are almost indistinguishable. Finally, the conclusion is a general summary, and not really related to the results presented in the note - it could have been written before anything else in the note.

Reply: The main purpose of this manuscript is to clarify the relationship and difference between these two models. The essential difference between these two models lies in the identification of the initial state. In the DLCM, the initial state is estimated, while that in the GNM is observed. As a result, the DLCM will have multiple solutions to the Nash cascade model, and the GNM can provide the unique result. For a long-time simulation in a linear system, the influence of the initial state can be ignored, as shown in Fig.1. However, in the real-time forecasting, the updated precision of initial state will have a great impact on the following predicted one, just as the first few predictions of the hydrographs in Fig.1. The conclusion has been rewritten as follows:

*Both the DLCM and GNM are derived from the Nash cascade model with a non-zero initial condition. The DLCM formulated the Nash cascade model in a matrix form based on the principles of state space analysis, while the GNM was written in a simpler algebraic expression after the complicated theoretical derivation. To clarify these two*

*models, the relationship and difference have been investigated mathematically and experimentally. The main conclusions are summarized as follows:*

*(1) The DLCM can be transformed to the GNM when the initial storage state is directly calculated by the linear storage-outflow relationship suggested in the Nash cascade model.*

*(2) The essential difference between these two models lies in the identification of the initial state. In the DLCM, the initial state is estimated, while that in the GNM is observed.*

*(3) The DLCM is an approximate solution of the Nash cascade model but not the exact solution due to its nonuniqueness of the initial estimated state. The GNM is the unique analytical solution of the Nash cascade model, whose initial state is implicitly written in a form of derivative and does not need to be estimated separately.*

---

## Author Comment (AC3) · 21 Oct 2019

**Responses to Anonymous Referee #2**

The study is based on the comparison of two well-known models. It seems that the objective is truly very narrow and does not contribute significantly to enhancing our understanding of hydrological system functioning. As such, I find it of very limited relevance to HESS. It might be more suitable to an application- or mathematically oriented Journal, even as I am not sure that there are significant advancements in terms of mathematical developments.

Reply: We thank Referee #2 for his/her suggestions and comments, which helped improving the manuscript. The Nash cascade model is a widely used flow routing model in hydrology. To make a higher precision of the model, many hydrologists have made a great effort to improve the model in different ways. The discrete linear cascade model (DLCM) and the generalized Nash model (GNM) are such models to improve the original Nash cascade model by considering the initial state. Based on the physical interpretation of the GNM, with the initial state included, the outflow can be thought to be generated by two parts, one is the initial storage water stored in river or watershed, and the other is the input (upstream inflow or precipitation). In the comparison of these two models, the reconstruction as well as the interpretation of the DLCM was made, which makes it more conceptual in hydrology and not only a mathematical formulation any more. We hope this comparison can help the model users make a clearer understanding of these two models and also the process of the outflow generation. The goal of this technical note is not only about the solution to a mathematical problem, but also about the physical interpretation of the flow routing process, which the hydrologists concern more.

A point which is not entirely clear to me is the basis for statements of the kind "The results show that the GNM provides a unique solution while the DLCM has multiple solutions depending on the estimate accuracy of the current state". They then state that observed values do not need to be estimated, thus implying that observed values are not associated with uncertainties. If a key difference is related to GNM being associated with observations (and not estimates), one could also claim that observations are always associated with measurement uncertainty/error. How do the Authors reconcile this aspect? There seem to be no mention of this aspect in the study.

Reply: This sentence is not clear and we have rewritten it as follows:

*The results show that the GNM provides a unique solution while the DLCM has multiple solutions, whose forecast precision depends upon the estimate accuracy of the current state.*

There seems to be a misunderstanding of the Referee #2 to our manuscript. The main purpose of this paper is to clarify the relationship and difference not the uncertainty analysis for these two models. So the measurement errors are not considered in this

paper. In hydrologic modeling, uncertainties can be classified into three primary types: structural errors, parameter errors, and data errors. Even for the uncertainty analysis, the difference between these two models is mainly from the model structure.

In the example section, the Authors mention relying on an optimization approach to estimate model parameters. It seems to me that parameter estimation uncertainty is neither quantified nor considered and I am not sure why.

Reply: As we have claimed, the purpose of this paper is to clarify the relationship and difference not the uncertainty analysis for these two models. So the parameter uncertainty is also not considered.

With reference to non-uniqueness of the solution, I am not sure why this is not compatible with typical uncertainty propagation analyses that are performed in environmental systems. Since there is uncertainty in some quantities, the latter should propagate to model outputs. Such an uncertainty can also be associated with initial conditions. The Authors should also comment on these aspects in future works.

Reply: The uncertainty propagation analysis is a good idea. With the initial state included in the two models, especially for GNM with an explicit expression of the initial state, uncertainty propagation of the initial state we think can be easily analyzed by Monte Carlo simulation. We will make such study in our future works. Thanks for the Referee's suggestion.

In terms of comparisons, I am not sure about the point raised by the Authors. They claim that the results obtained by the DLCM approach are approximated (I guess when considering results obtained through the continuous counterpart of an otherwise discretely sampled signal). I am not sure about what elements we learn from this exercise with respect to other studies on signal analysis that are available in the literature.

Reply: There seems to be a misunderstanding of the Referee #2 to this conclusion. Actually, all the models including the DLCM and GNM in this study are approximations to the real world. In our conclusion, the DLCM is said to be an approximation of the Nash cascade model mathematically not because it is sample-data based but because its initial state is estimated. In comparison, the initial state in the GNM is implicitly written in a form of derivative strictly based on the linear reservoir assumption and does not need to be estimated separately. Hence, the GNM is an exact solution to the Nash cascade model.

Finally, it should be noted that the quality of the English is really substandard, thus posing difficulties to the reviewer. I am not providing specific examples simply because they are widespread throughout the text.

Reply: The manuscript was carefully reread to check for language issues. We have replaced the initial mistakes and edited the sentences carefully. Thanks again!